# Roles of the INO80 and SWR1 Chromatin Remodeling Complexes in Plants

**DOI:** 10.3390/ijms20184591

**Published:** 2019-09-17

**Authors:** Jianhao Wang, Sujuan Gao, Xiuling Peng, Keqiang Wu, Songguang Yang

**Affiliations:** 1Key Laboratory of South China Agricultural Plant Molecular Analysis and Genetic Improvement, Guangdong Provincial Key Laboratory of Applied Botany, South China Botanical Garden, Chinese Academy of Sciences, Guangzhou 510650, China; xn2112wjh@163.com (J.W.); m13203206354_2@163.com (X.P.); 2University of Chinese Academy of Sciences, Chinese Academy of Sciences, Beijing 100049, China; 3College of Light Industry and Food Science, Zhongkai University of Agriculture and Engineering, Guangzhou 510225, China; gaoshj@126.com; 4Institute of Plant Biology, National Taiwan University, Taipei 106, Taiwan

**Keywords:** chromatin remodeling, INO80/SWR1 complexes, NuA4 complex, histone variant H2A.Z, gene regulation, plant development

## Abstract

Eukaryotic genes are packed into a dynamic but stable nucleoprotein structure called chromatin. Chromatin-remodeling and modifying complexes generate a dynamic chromatin environment that ensures appropriate DNA processing and metabolism in various processes such as gene expression, as well as DNA replication, repair, and recombination. The INO80 and SWR1 chromatin remodeling complexes (INO80-c and SWR1-c) are ATP-dependent complexes that modulate the incorporation of the histone variant H2A.Z into nucleosomes, which is a critical step in eukaryotic gene regulation. Although SWR1-c has been identified in plants, plant INO80-c has not been successfully isolated and characterized. In this review, we will focus on the functions of the SWR1-c and putative INO80-c (SWR1/INO80-c) multi-subunits and multifunctional complexes in *Arabidopsis thaliana*. We will describe the subunit compositions of the SWR1/INO80-c and the recent findings from the standpoint of each subunit and discuss their involvement in regulating development and environmental responses in *Arabidopsis*.

## 1. Introduction

Eukaryotic DNA is packaged with histones to form an inherently stable complex structure known as chromatin. The basic repeating unit of chromatin is the nucleosome, which consists of approximately 146 base pairs of DNA wrapped on a histone octamer containing two molecules each of histone H2A, H2B, H3, and H4 [1]. Therefore, to allow DNA processing, such as DNA replication, repair, and recombination, a dynamic chromatin environment is needed. Two main players, histone post-translational modifications by histone-modifying enzymes and nucleosome positioning by ATP-dependent chromatin-remodeling complexes (CRCs) are involved in the regulation of chromatin dynamics [2,3]. In general, CRCs can destabilize the interactions between histone octamers and DNA by using the energy derived from ATP hydrolysis [4], thus increasing the binding efficiency of transcription factors.

The catalytic subunit (ATPase) of CRCs belongs to the SWI/SNF family [5], which is part of a large superfamily of helicases and translocases called superfamily 2 (SF2). The SWI/SNF family is named after the first identified CRC from *S. cerevisiae* by the examination of mating type switching (*SWI*) and sucrose nonfermenting (*SNF*) mutants [6]. Further studies conducted with *Drosophila*, mammals, and plants revealed that CRCs widely exist in eukaryotes. Based on the evolutionarily conserved central ATPase subunit, CRCs are currently classified into four subfamilies: SWI/SNF, ISWI (imitation SWI), INO80 (Inositol Requiring 80) and CHD (chromo and DNA-binding domain) [7]. CRCs of the INO80 family include the yeast INO80 complex (INO80-c) and its orthologues in humans and plants, the yeast SWR1 (SWi2/snf2-Related 1) complex (SWR1-c) and its orthologues SRCAP (SNF2-RELATED CBP ACTIVATOR PROTEIN) (human) and SWR1 (*Arabidopsis*) (Table 1). Among those families, SWR1/INO80-c plays essential roles in DNA repair, checkpoint regulation, DNA replication, telomere maintenance and chromosome segregation [8,9].

Although the molecular mechanisms of SWR1/INO80-c are largely unknown, a large body of genetic and biochemical evidence strongly suggests that SWR1/INO80-c are also present in plants. In this review, we compare the functions and subunit compositions of the SWR1/INO80-c of plants and other organisms including yeasts and mammals. We also discuss the recent investigations underpinning the involvement of SWR1/INO80-c subunits in regulating key plant growth and development processes, as well as environmental responses.

## 2. Chromatin Remodeling Functions of INO80 and SWR1 CRCs

Biochemical analyses indicated that SWR1-c promotes the replacement of nucleosomal H2A/H2B dimers with H2A.Z/H2B [10], while INO80-c catalyzes the reverse dimer exchange reaction [11,12,13]. Further studies demonstrated that H3-K56Ac (H3K56 acetylation) or H3-K56Q (substitution H3-K56 for glutamine) switches the substrate specificity of SWR1-c, inhibiting H2A.Z deposition [13]. Consistently, it was observed that H2A.Z deposition is inhibited by H3-K56Q but not by SWR1-c or INO80-c [14]. However, recent direct EMSA assays indicated that INO80-c catalyzes the replacement of H2A.Z/H2B with H2A/H2B on an H2A.Z/H3-K56Q substrate [15]. The structural analysis of the core INO80-c from the fungus *Chaetomium thermophilum* showed that the two AAA^+^ ATPases, Rvb1 (RuvB-like protein 1) and Rvb2, form heterohexamers, acting as a ‘stator’ for the motor (ATPase) and nucleosome-gripping subunits [16]. The ATPase of INO80-c binds to nucleosomal DNA at superhelical location 6 (SHL6), unwraps approximately 15 base pairs, disrupts the H2A–DNA contacts and is poised to pump entry DNA into the nucleosome [16]. Arp5 (Actin-Related Protein 5)–Ies6 (Ino Eighty Subunit 6) binds at SHL2 and SHL 3 acting as a counter grip for the motor on the other side of the H2A–H2B dimer. On the other hand, yeast SWR1-c is also assembled around a heterohexameric core of the Rvb1 and Rvb2 subunits, and the motor subunit binds at SHL2, a position it shares in common with other remodelers [17].

In yeast, the histone variant H2A.Z (Htz1) is enriched in intergenic regions compared with coding regions, and it is involved in the activation of a subset of genes [18]. Meanwhile, incorporation of Htz1 into nucleosomes influences histone modifications and chromatin remodeling, supporting a role for Htz1 at inactive promoters [18]. In *Arabidopsis*, the high-resolution ChIP-exo assay showed that SWR1 and INO80-c are targeted selectively to the +1 nucleosome of essentially all genes via their subunits [19]. For instance, Swc4 (SWR complex subunit 4) and Swc5 occupy nucleosome positions in the body of genes in addition to +1, while Rvb1 and Swc2 have correlated co-occupancy levels that span a wide range over different promoters [19]. Interestingly, a previous study showed that SWR1/INO80-c may be involved in DNA methylation, since a strong anti-correlation between DNA methylation and H2A.Z deposition was observed [20]. However, the H2A.Z deposition at gene bodies is correlated with a lower transcription level, presumably by destabilizing constitutive gene expression rather than promoting gene body DNA methylation [21]. The genomic binding maps of the subunits of SWR1/INO80-c suggested that many subunits have a rather plastic organization that allows their subunits to exchange between the two complexes [19].

## 3. The Subunits of INO80 and SWR1 CRCs

yIno80, the ATPase of yeast INO80 complex, was identified in a screen for regulators of phospholipid biosynthesis [22]. Further studies found that yIno80 associates with 14 other proteins including Rvb1, Rvb2, Arp4, Arp5, Arp8, actin1, Taf14 (TATA binding protein-Associated Factor 14), Ies1-6, and Nhp10 (Non-Histone Protein 10) to form a complex (INO80 complex) of 1.2 Mda [22,23] (Table 1). ySwr1, the homolog of yIno80, was subsequently identified and found to catalyze the exchange of H2A for Htz1 [10,24]. Proteomic data indicated that SWR1-c consists of 14 subunits sharing 4 components (Rvb1, Rvb2, Arp4, Act1) with the INO80 complex (Table 1). Interestingly, 4 subunits (Arp4, Yaf9-YEAST ALL1-FUSED GENE FROM CHROMOSOME 9, Swc4, and Act1) of SWR1-c are also part of the NuA4 (Nucleosome Acetyltransferase of H4) complex that is a key regulator of transcription, cellular response to DNA damage and cell cycle control [25,26]. In addition to acetylation of histone H4, histone H2A and histone H2A variant Htz1 [27,28,29], the yeast NuA4 complex also targets non-histone proteins, which encompass metabolism, RNA processing and stress response [30]. In human, Tip60 (Tat-interactive protein 60), the homolog of acetyltransferase subunit of yNuA4 complex (Esa1, Essential for *SAS* Family Acetyltransferase), was originally identified as an interacting partner of the HIV1-tat protein [31]. Subsequently, the purified Tip60 complex contains at least 16 subunits harboring HAT activity towards histone H4, H2A and H2A.Z in chromatin [32,33,34]. The subunit composition analysis indicates that the Tip60 (NuA4) complex is a fusion form of yeast NuA4 and SWR1 complexes since it also has the enzymatic activity of ATP-dependent H2A.Z-H2B histone dimer exchange [35] (Table 1). A recent report showed that a dynamic merge and separation of NuA4 and SWR1 complexes control cell fate plasticity in *C. albicans* [36]. Indeed, the YEATS domain of Yaf9 can recognize the acetylation of Eaf1 K173 and mediate the Yaf9-Eaf1 interaction. Furthermore, the reversible acetylation and deacetylation of Eaf1 by Esa1 and Hda1 control the merge and separation of NuA4 and SWR1 complexes, and this regulation is triggered by Brg1 recruitment of Hda1 to chromatin in response to nutritional signals that sustain hyphal elongation [36].

To date, the structural insights into the plant SWR1/INO80-c are largely unknown. Based on sequence homology and protein-protein interacting data, the possible subunits orthologues of yeast and human SWR1/INO80-c were identified in *Arabidopsis* (Table 1). The plant ATPase of SWR1-c, PHOTOPERIOD-INDEPENDENT EARLY FLOWERING 1 (PIE1), was first characterized as an activator of *FLOWERING LOCUS C* (*FLC*) in flowering [37]. Further affinity purification followed by tandem mass spectrometry (TAP-MS) assays showed that SWR1-c contains at least 12 subunits, including RVB2A/B, YAF9A/B, and SWC6 in *Arabidopsis* [38]. Furthermore, a more recent study identified 11 conserved SWR1-c subunits in *Arabidopsis* using the conserved SWR1 subunit ARP6 as a bait in tandem affinity purification experiments [39]. Unlike SWR1-c, the compositions of *Arabidopsis* INO80-c are based on the sequence homology and protein-protein interacting data. The homologue of yINO80 was found in *Arabidopsis* as a factor in homologous recombination [40]. These data suggested that plants may also have the INO80 complex. Interestingly, recent studies indicated that the NuA4 complex may exist in *Arabidopsis*, since YAF9A, the yYAF9 homologue (a subunit of yeast NuA4 complex), targets to *FLC* chromatin and regulates the acetylation of H2A.Z and H4 [41]. The multiplication of the subunits in yeast, human and plant suggest that INO80/SWR1 CRCs are conserved in eukaryotes.

## 4. Involvement of INO80/SWR1-c in DNA Repair

In yeast, the *ino80* and *swr1* mutants are hypersensitive to DNA damage-inducing agents [42,43], suggesting a direct role for INO80 and SWR1 CRCs in DNA repair. Indeed, *ino80* mutants have defects in both the homologous recombination (HR) and nonhomologous end-joining (NHEJ), two pathways for the repair of DNA double-strand breaks (DSBs), while *swr1* mutants have defects in NHEJ alone [44]. Similarly, *Arabidopsis INO80* also plays a role in DSBs (Table 2, Figure 1). Under standard growth conditions, the *ino80* mutants show a reduction of the HR frequency compared with wild-type plants [40,45]; however, after genotoxic treatment, HR in the mutant increased, accompanied by more DNA double-strand breaks and stronger cellular responses [45]. Further analyses show that *INO80* promotes HR downstream of the chaperone NRP1 (NAP1-RELATED PROTEIN 1) after the formation of γ-H2A.X foci during DNA damage repair [46]. Nevertheless, the functional coordination of *INO80* and *NRP1/2* was also observed in apical meristems during plant growth and development [47]. In addition to INO80, ARP5, the conserved subunit of the INO80-c, also plays key roles in the DNA repair. Indeed, ARPs are highly similar to actin, but they cannot polymerize and do not have ATPase activity. Currently, ARPs are classified into 11 subfamilies, and ARP4-ARP9 are predominantly localized in the nucleus [48]. As dedicated conserved subunits of SWI/SNF and INO80 CRCs, ARPs associate directly with ATPases via the conserved N-terminal HSA (helicase-SANT-associated) domain [49]. The *arp5* plants were hypersensitive to DNA-damaging reagents [50], and the transcription levels of DNA repair genes *RAD51*/*RAD54* were up-regulation in mutant plants [51].

Similar to *INO80* and *ARP5*, SWR1-c is also important for DNA repair in *Arabidopsis*. Since mutations in genes for SWR1-c subunits *PIE1*, *ARP6* and *SWC6* cause hypersensitivity to various DNA damaging agents [68]. The reduced DNA repair capacity of those mutants is connected with impaired HR, in contrast with the hyper-recombinogenic phenotype of yeast SWR1 mutants [44,68]. This suggests functional diversification of SWR1-c between lower and higher eukaryotes. The *arp5 arp6* double mutants displayed a higher percentage of DNA in tails than that of individual single mutants, suggesting that function of ARP5 in the INO80-c acts independently and/or synergistically with the ARP6-containing SWR1-c in DNA repair [51].

## 5. Functions of INO80/SWR1-c in Flowering

Successful reproduction in plants requires the coordinated transition from vegetative growth to reproductive development. Complex and intricate gene-regulatory networks of transcription factors and chromatin remodelers guide flowering time and flower development, while integrating both internal and external signals [75]. In *Arabidopsis*, two MADS box transcription factors *MADS AFFECTING FLOWERING 4/5* (*MAF4/5*), *FLC* and *SHORT VEGETATIVE PHASE* (*SVP*) act as negative regulators of flowering time via directly repressing the expression of the floral pathway integrators *FLOWERING LOCUS T* (*FT*) and *SUPPRESSOR OF OVEREXPRESSION OF CONSTANS 1* (*SOC1*) [76,77,78].

The core subunit components of the *Arabidopsis* SWR1-c, including PIE1, ARP6, ARP4, SWC6/ SERRATED LEAVES AND EARLY FLOWERING (SEF), SWC4 and YAF9A/B have been shown to play important roles in regulating the proper growth and flowering. The *Arabidopsis* ATPase of SWR1-c, PIE1, was first identified as an activator of *FLC* [37]. Mutations in *PIE1* repress the *FLC*-mediated delay of flowering, as a result of the presence of *FRIGIDA* [37]. Further research demonstrated that PIE1 associates with the SWR1-c subunits ARP6 (also known as EARLY IN SHORT DAYS 1 [ESD1] and SUPPRESSOR OF FRI 3 [SUF3]) and SWC6 in repression of flowering [53,60,61,66,67]. Indeed, loss of H2A.Z from chromatin in *arp6* and *pie1* mutants results in reduced *FLC* expression and premature flowering, indicating that H2A.Z is required for the expression of *FLC* [62], which suggested a link between PIE1 and H2A.Z incorporation.

Next to ARP6, ARP4 (present in both INO80-c and SWR1-c, Table 1) has also been shown to act in plant reproductive development. Knockdown of *ARP4* results in strong pleiotropic phenotypes such as altered organization of plant organs, early flowering, delayed flower senescence and high levels of sterility, indicating its important roles in plant development [59]. Besides, TAP-MS assays showed that the multiple subunits of SWR1, NuA4, INO80 and SWI/SNF complexes copurified with ARP4, suggesting that ARP4 may be a core subunit of those complexes [56].

In yeast, Swc2 together with Swc6 and Arp6 forms SWR1 complex 2, which handles binding to the histone variant H2A.Z associated with the nucleosome and facilitating histone exchange [79]. To date, three SWCs including SWC2, SWC4 and SWC6 were found in *Arabidopsis*. *SWC4*-deficient mutants are embryo-lethal, while *SWC4* knockdown lines display acceleration of flowering time, indicating that SWC4 controls post-embryonic processes [38]. However, unlike PIE1, SWC4 represses flowering via repressing *FT* [38]. Indeed, SWC4 mediates the recruitment of SWR1-c to the target chromatin regions through recognizing AT-rich DNA elements that are over-represented in the promoters of a subset of genes (*FT*, *FUL*, *IAA19* and *ERF9*) where H2A.Z incorporation impairs transcription [38]. Intriguingly, like yeast SWC4, recent data demonstrated that SWC4 also associates with YAF9s in *Arabidopsis* by TAP-MS assays [41]. *SWC6* was originally identified from *sef* mutants. The *sef* plants present a pleiotropic phenotype including serrated leaves, frequent absence of inflorescence internodes, bushy aspect, and early flowering [53,60]. Like PIE1, SWC6 also activates the transcription of *FLC* via interacting with SUF4 and FLX, two subunits of the FRIGIDA transcription activator complex [52]. Meanwhile, defects in the splicing of *SWC6* pre-mRNA mediated by SKIP (SNW/SKI-INTERACTING PROTEIN) reduce H2A.Z enrichment at *FLC*, *MAF4*, and *MAF5,* resulting in reduced expression of these genes [80]. Interestingly, SWC6 interacts with HAM1 [38], a Myst family histone acetyltransferase of the putative *Arabidopsis* NuA4 complex (NuA4-c) [81]. Although yeast two-hybrid assays showed that SWC2 interacts with H2A.Z and SWC6, the function of *SWC2* in *Arabidopsis* has not been characterized so far [61].

Yeast Yaf9 (YEAST ALL1-FUSED GENE FROM CHROMOSOME 9) contains an N-terminal YEATS domain and a C-terminal predicted coiled-coil sequence [82]. The YEATS domain proteins are conserved from yeast to human, and function as transcriptional regulators as a part of chromatin-modifying complexes [83]. Indeed, as a selective reader of H3K27ac, the YEATS domain of Yaf9 is essential for deposition of H2A.Z, gene transcription and the DNA-damage response [84]. Deletion of *Yaf9* shows reduced telomere-proximal gene expression and sensitivity to DNA-damaging agents [85]. Additionally, *Yaf9* has involved chromosome segregation, telomere silencing and response to spindle stress [82]. Moreover, SWR1-c and NuA4-c share 4 subunits (Act1, Arp4, Swc4, Yaf9), indicating a possible functional link between these two complexes in yeast [86]. The *Arabidopsis* Yaf9 homolog YAF9A acts as a negative regulator of flowering by controlling the H4 acetylation levels in *FLC* and *FT* chromatin [41,73]. Furthermore, CCA1 (CIRCADIAN CLOCK ASSOCIATED1) recruits YAF9A to the promoter region of *GI* (*GIGANTEA*), resulting in accumulation of H2A.Z and the acetylation of H4, thus increasing *GI* expression [74]. Indeed, recent data demonstrated that YAF9A and YAF9B have unequally redundant functions, and they regulate flowering time by both *FLC*-dependent and independent mechanisms [41]. Moreover, YAF9A interacts with HAM1, a putative catalytic subunit of NuA4-c, causing high level acetylation of H2A.Z and H4 of *FLC* chromatin without affecting the deposition of H2A.Z at the *FLC* locus [41]. YAF9 proteins also associate with other speculative SWR1-c subunits in *Arabidopsis* [38,56]. Together, these data suggest functional conservation of the SWR1-c-NuA4-c module in plants and yeast. In addition to flowering, SWR1-c subunit loss-of-function mutants also exhibit pleiotropic flower phenotypes including frequent absence of inflorescence internodes, bushy growth and flowers with altered organ number and size [37,60,62,66,69].

Unlike the SWR1-c, *Arabidopsis* INO80-c plays a positive role in flowering. INO80, the ATPase of INO80-c is involved in flowering time control via interacting with H2A.Z. Indeed, INO80 binds within the gene body, enhances H2A.Z enrichment and maintains low expression levels of the key flowering repressor genes *FLC* and *MAF4/5*, thus *ino80* mutants displaying late flowering phenotype [45]. Recent analysis demonstrated that INO80 interacts with ARP5 and acts in concert with ARP5 during plant cellular proliferation and replication stress response [51]. Nevertheless, ARP5 is not required for INO80-mediated control of flowering time and related transcriptional regulation of *FLC* and *MAF4*/*MAF5* [51]. Taken together, these data indicated that SWR1-c and INO80-c may play an opposite role in flowering, which is consistent with the biochemical activity of SWR1-c and INO80-c.

## 6. Functions of INO80/SWR1-c in Immunity Response

Plants need to precisely coordinate defense with growth and development to optimize environmental conditions. A previous study showed that 65% of the differentially expressed genes in *hta9 hta11* mutants (loss of two major H2A.Z encoding genes out of the possible three in *Arabidopsis*) were also misregulated in *pie1* plants, and the majority of misregulated genes were related to systemic acquired resistance (SAR) [63]. These data indicated that SWR1-c plays a role in maintaining a repressive state of the SAR response. However, further phenotypic analysis showed that *pie1*, *swc6*, and *hta9 hta11* mutants displayed more macroscopic disease symptoms and increased susceptibility toward *Pst* DC3000 compared with the wild-type, whereas the *arp6* mutant showed increased resistance [57]. These observations suggest that H2A.Z, PIE1, and SWC6 are essential for basal resistance in *Arabidopsis*, whereas ARP6 has an opposite function. Moreover, although the *arp6* and *swc6* mutants have similar morphological and developmental phenotypes (early flowering and serrated leaves), and ARP6 and SWC6 have been shown to physically interact in *Arabidopsis* [53,60], the RNA-seq data did not show a strong correlation in these mutants [57]. This observation is consistent with the defense phenotypes, suggesting that ARP6 and SWC6 could have distinct functions in *Arabidopsis*.

To data, little is known about INO80-c in plant immunity response. However, *Arabidopsis* RVB1/RIN1 (RESISTANCE TO PSEUDOMONAS SYRINGAE PV MACULICOLA INTERACTOR 1), the common subunit of INO80/SWR1-c, interacts with both RPM1 (CC-NB-LRR) and RPP5 (TIR-NB-LRR) proteins to negatively regulate the expression of disease resistance (R) genes [58]. Indeed, RVB1 belongs to the members of a larger family of proteins known as the AAA^+^ class chaperone-like ATPases [87]. In bacteria, Rvb1 and Rvb2 form a double hexamer around Holliday junctions to promote their migration during homologous recombination in DSB repair [88]. In yeast, Rvb1 and Rvb2 are present in both SWR1/INO80-c and are homologous to *Arabidopsis* RVB1and RVB2A/B. *RVB1* is essential for *Arabidopsis* development during both female gametophyte and sporophyte development [58]. Meanwhile, ectopic expression of *RVB1* in yeast *yRvb1* deletion alleles can rescue yeast viability [58]. Intriguingly, although TAP-MS experiments showed that RVB1 may not be the subunit of *Arabidopsis* SWR1-c [39], the interaction between RVB1 and YAF9A still suggested that RVB1 associates with SWR1-c [41]. Compared with yeast, little is known about RVB2A/B functions in plants. Recent data showed that RVB2A/B physically interact with the EAF1 subunit of the NuA4 complex [56], which is a homolog of yeast Eaf1. Further research is required to investigate the molecular mechanism of how INO80-c is involved in plant defense.

## 7. Involvement of INO80/SWR1-c in the Regulation of MicroRNA Expression

MicroRNAs (miRNAs) repress protein production post-transcriptionally. In flowering plants, modules of miRNAs and their target transcription factors, such as the miR156-SPLs/miR172-AP2 LIKEs and miR319-TCPs/miR164-CUCs modules, control diverse developmental processes including phase transitions, leaf shape, and floral organ identity [89,90]. Previous studies suggest that SWR1-c contributes to the fine control of plant development by generating a balance between miRNAs and target mRNAs at the transcriptional level. In the mutants of *arp6*, *sef*, and *pie1*, miR156 and miR164 levels are reduced at the transcriptional level, which results in the accumulation of target mRNAs and associated morphological changes [54]. Further investigation showed that *arp6*, *sef*, and *hta9*
*hta11* mutant plants reduce the expression of *MIR156A* and *MIR156C* via facilitating the deposition of H3K4me3, rather than decreasing nucleosome occupancy [55].

The functions of INO80-c in miRNA regulation are largely unknown. However, in mammals, *INO80* is a target gene of miR148a in the cancer stem cells (CSCs) of anaplastic thyroid carcinoma (ATC) [91]. The expression of *INO80* was upregulated in ATC-CSCs and downregulated upon miRNA148 overexpression. Moreover, overexpression of miRNA148a and knockdown of *INO80* acted synergistically to decrease the expression of stem cell marker genes as well as to attenuate stem cell-specific properties including the ability to form tumors [91]. These results suggested that in addition to miRNA regulation, miRNA may also affect the functions of INO80-c or SWR1-c.

## 8. Other Functions of Core Subunits of INO80/SWR1-c

In addition to the functions of INO80/SWR1-c described above, the core subunits of INO80/SWR1-c have other roles in plants development. For instance, the accumulation of anthocyanin in H2A.Z deposition-deficient mutants such as *pie1* is associated with increased H3K4me3 of anthocyanin biosynthetic genes, which is negatively associated with the presence of H2A.Z [64]. These results reveal an antagonistic relationship between H2A.Z and H3K4me3 in the regulation of anthocyanin biosynthesis genes. However, genome-wide occupancy assays of H2A.Z, H3K4me3 and H3K27me3 showed that H2A.Z preferentially associates with H3K4me3 at promoters and H3K27me3 at enhancers in *Arabidopsis* inflorescence. Moreover, H2A.Z represses enhancer activity by promoting H3K27me3 and preventing H3K4me3 histone modifications [92]. A recent study demonstrated that H3K27me3-enriched chromatin is dependent on the prior action of PIE1 and H2A.Z. Indeed, the H2A.Z- and H3K27me3-enriched chromatin is subsequently stabilized during transcription by the CHD remodeler PICKLE and the histone methyltransferase CURLY LEAF [65]. Collectively, these data suggested that there is a cross-talk between H2A.Z deposition and histone modifications.

Moreover, *ARP6* also regulates the female meiosis of prophase I by activating the expression of meiotic recombination related genes such as *DMC1* (*DISRUPTED MEIOTIC cDNA1*) [69]. Subsequent research revealed that ARP6 associates with SWR1-c and promotes the expression of *WRKY28* through H2A.Z deposition in megaspore mother cells (MMC) [70]. Intriguingly, *arp6* null alleles exhibit a delayed germination rate in osmotic stress conditions, suggesting a positive role of ARP6 in osmotic stress responses [71]. Meanwhile, *arp6* plants also display an apparent phosphate starvation response (PSR), when grown in a phosphate replete medium, since ARP6 is required for proper H2A.Z deposition at a number of *PSR* genes [72]. Moreover, overexpressing *ARP6s* from other plant species such as *Physcomitrella patens* and rice (*Oryza sativa*) could rescue the early flowering phenotype of *Arabidopsis arp6* mutants, suggesting that the function of ARP6 is conserved in plants [93].

## 9. Concluding Remarks and Future Perspectives

The roles of SWR1/INO80-c in nuclear activities are quite diverse ranging from DSB repair to the regulation of gene expression. The mutants of *INO80-c* and *SWR1-c* subunits display a variety of organism-specific phenotypes. Moreover, phenotypic differences among the single and double mutants of *INO80-c* and *SWR1-c* subunit genes are also observed. For instance, the phenotype of *pie1* plants is more severe than that of other SWR1 component mutants, suggesting that PIE1 may be a scaffolding component of different complexes. Indeed, like human p400, PIE1 contains HSA, ATPase, and SANT domains, which are found separately in the Swr1 and Eaf1 proteins, implying that PIE1 may also be an ortholog of p400. TAP-MS data showed that *Arabidopsis* TRA1 (also a subunit of Spt-Ada-Gcn5-acetyltransferase [SAGA] complex), the homolog of the yeast NuA4 subunit Tra1 and the mammalian Tip 60 subunit TRRAP, also presents in SWR1-c. Collectively, these data suggest that the same subunit (like PIE1) may exist in different distinct complexes. The SWR1-c components also play different roles in resistance to different pathogens. For instance, *pie1* and *swc6* plants display reduced basal resistance, while loss of *ARP6* fucntion results in enhanced resistance [57]. Further in-depth analyses are needed to uncover the functional specificity of SWR1/INO80-c and NuA4-c subunits and their interaction in plants.

Although SWR1/INO80-c are associated with histone variants in vivo [45,63] (Table 2; Figure 1), the precise chromatin-remodeling mechanisms of these complexes in diverse cellular processes are largely unknown. For instances, the mechanism of how SWR1/INO80-c are recruited to their target loci such as *FLC* is still unclear [37,45,60,61]. It is possible that there is a cross-talk between chromatin modifications and histone variant exchange. In yeast, the NuA4-c mediated acetylation of specific histones in the nucleosome is important for SWR1-c targeting to chromatin and H2A.Z incorporation [28,94]. Moreover, it was shown that Bdf1, a bromodomain-containing subunit of yeast SWR1-c, recruits the complex to chromatin by recognizing acetylated H4 tails [28,95]. Consistently, loss-of-function of Bdf1 results in global reduction of H2A.Z in chromatin [96]. Indeed, the Yaf9 subunit of SWR1-c (also a subunit of NuA4-c) targets histone H3K27ac through its YEATS domain in yeast [84]. The *Arabidopsis* Yaf9 homologs YAF9A and YAF9B are also novel histone readers that bind to unmodified and acetylated histone H3 [41]. Meanwhile, MBD9, a bromodomain and homeodomains (PHD)-containing unique subunit of *Arabidopsis* SWR1-c, also recognizes the acetylated histones [97]. Further studies demonstrated that in *mbd9* plants, the level of H2A.Z incorporation is significantly reduced, and H2A.Z sites depending on MBD9 has higher levels of H3K9Ac and lower levels of H3K4me3 and H3K36me3 [39]. These data suggest that MBD9, like Bdf1 in yeast, could target SWR1-c via its bromodomain by recognizing acetylated histone marks, such asH3K9Ac. Further research is required to investigate whether acetylated residues on histones can directly recruit SWR1-c to facilitate distinct nuclear processes. In addition, genome-wide binding analyses of INO80/ SWR1-c combined with transcriptome analysis will be useful for the identification of the direct target genes regulated by SWR1/INO80-c, which may provide a comprehensive understanding of how SWR1/INO80-c control plant development and stress responses.

## Figures and Tables

**Figure 1 ijms-20-04591-f001:**
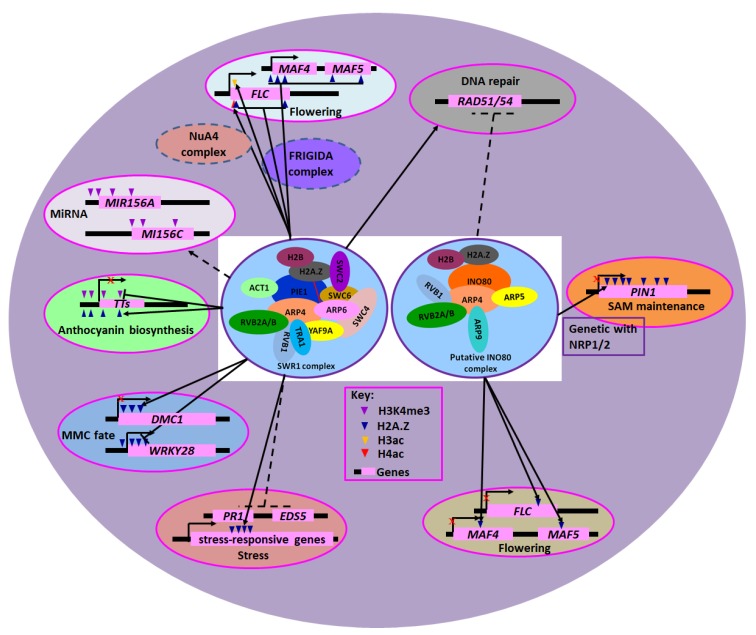
INO80 and SWR1 Chromatin Remodeling Complexes in Cellular Processes and Plant Development. The compositions of putative INO80 and SWR1 complex are indicated. The purple oval represents the nucleus of plant cells. The overlapped ovals and red lines represent interacting subunits of the INO80 and SWR1 complex. Lines with perpendicular bars and red cross denote repression, while arrows indicate enhancement or activation. Broken lines indicate relationships not proven to be direct.

**Table 1 ijms-20-04591-t001:** Compositions of INO80, SWR1 and NuA4 Complexes in *Arabidopsis.*

Complex									
Organism	Yeast			Human			*Arabidopsis*		
Family and Composition	INO80	SWR1	NuA4	INO80	SWR1/SRCAP	NuA4/Tip60	INO80	SWR1	NuA4
ATPase or Acetyltransferase	Ino80	Swr1	Eaf1 **, Esa1 *	hIno80	SRCAP	p400 **, Tip60 *	INO80	PIE1	HAM1/2 *, EAF1 **
Noncatalytic homologous subunits	Rvb1, Rvb2			Tip49a, Tip49b			RVB1/RIN1		
							RVB2A, RVB2B		
	Arp4, Actin1			BAF53a			ARP4, ACT1		
	Arp5, Arp8	Arp6		Arp5, Arp8	Arp6	Actin	ARP5, ARP9	ARP6	
	Taf14	Yaf9			GAS41			GAS41/YAF9A, TAF14/YAF9B	
	Ies2, Ies6			hIes2, hIes6					
		Swc4/Eaf2			DMAP1			SWC4	
		Swc2/Vps72			YL-1			SWC2	
		Swc6/Vps71		ZnF-HIT1				SWC6	
		Bdf1			Brd8/TRCp120				
		H2A.Z, H2B			H2A.Z, H2B		H2A.Z, H2B		
			Tra1			TRRAP		TRA1	
			Epl1			EPC1			
			Yng2			ING3			ING1, ING2
			Eaf3			MRG15			MRG1, MRG2
			Eaf5						
			Eaf7			MRGBP			
			Eaf6			hEaf6			
Unique	Ies1, Ies3, Ies 4, Ies5, Nhp10	Swc3,5,7		Amida, NFRKB, MCRS1, FLJ90652, FLJ20309		MRGX, FLJ11730, MRGBP, EPC1, EPC-like		MBD9, AL5-7	

* and ** represent the acetyltransferase and ATPase subunits of NuA4-c, respectively. The subunits of INO80 and NuA4 complexes in *Arabidopsis* are based on the sequence homology and protein-protein interacting data.

**Table 2 ijms-20-04591-t002:** The Functions of core SWR1/INO80-c Subunits in *Arabidopsis.*

	Gene	Interacting Proteins *	Functions	Reference
The common subunits of INO80/SWR1-c	SWC2		unknown	
	SWC4		Flowering time control; Leaf cell proliferation and expansion	[38]
	SWC6/SEF	SUF4, FLX, TAF14	Flowering time control	[52,53]
			MicroRNA expression	[54,55]
		HAM1, EAF1		[38,56]
			Immunity response	[57]
	RVB1/RIN1	FLX, SUF4, FES1, FRI		[52]
		RPM1, RPP5	Sporophyte and female gametophyte; Disease resistance	[58]
	RVB2A, RVB2B	EAF1		[56]
	ARP4		Multiple effects on plant development	[59]
			Core subunit of SWR1, NuA4, INO80 and SWI/SNF complexes	[56]
Unique subunits of INO80-c	INO80		Controls homologous recombination	[40,45,46,51]
			Flowering time control	[45,51]
			Apical meristems maintenance	[47]
	ARP5		organ development; DNA repair	[50,51]
	ARP9		unknown	
Unique subunits of SWR1-c	PIE1		Flowering time control	[37,60,61,62]
			Immunity response	[57,63]
			MicroRNA expression	[54]
			Anthocyanin biosynthesis	[64]
			Maintenance of H3K27me3	[65]
	ARP6		Flowering time control	[61,62,66,67]
			MicroRNA expression	[54,55]
			Immunity response	[57]
			DNA repair	[68]
			Female meiosis regulation; Germ-line specification	[69,70]
			Osmotic stress; Phosphate starvation response	[71,72]
	YAF9A	CCA1, HAM1	Flowering time control	[41,73,74]
	YAF9B			

* The interactions among the core subunits of SWR1/INO80-c are shown in Figure 1.

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
