# Peer review of "Roles of the INO80 and SWR1 Chromatin Remodeling Complexes in Plants"

_ijms, 2019, doi:10.3390/ijms20184591_

Round 1

Reviewer 1 Report

The authors present a nice review in which they address the roles of the INO80 and SWR1 chromatin remodeling complexes in plants, and compare the functions and subunit compositions of the SWR1/INO80-c of plants, yeasts and mammals. They also discuss the involvement of SWR1/INO80-c subunits in regulating plant growth, development and environmental responses.

I consider that the scientific topic being addressed is appropriate and interesting. The paper is well written and clear in all its sections. Cited references are appropriate and adequate, listing a large number of publications in high-impact journals.

I recommend the publication of this paper in the International Journal of Molecular Sciences, and have very few minor comments and suggestions listed below:

The role of ARP6 in maintaining genome stability and in DNA repair has been omitted from Table 2. Regarding this function, reference Rosa et al. 2013 could be used, together with ref. [49] mentioned in line 154.

Li et al. (2005) have shown that in yeast the presence of Htz1 under the inactivated conditions is essential for optimal activation of a subset of genes, and suggest that Htz1 may serve to mark quiescent promoters for proper activation. They have also demonstrated that Htz1 is not generally responsible for nucleosome positioning, even at those promoters where he is highly enriched, and that incorporation of Htz1 into nucleosomes inhibits activities of histone modifiers associated with transcription.
If found convenient, this information could be used to supplement the information brought in lines 76 - 84.

The abbreviation description for FLC is brought in line 168, although the first appearance of this term is in line 116. In addition, some terms throughout the text appear without their abbreviation explanation, e.g. HSA (line 144), GI (line 218), etc. Due to the quite extensive number of abbreviations used in the text, maybe abbreviation list could be provided at the end of the manuscript, if possible.

Term Htz1 (used in line 93 and 98) has not been brought into association with H2A.Z in the manuscript.

At the end of the line 91, authors could refer also to Table 1, in addition to the existing references.

Figure numeration is missing from the Figure 1 description.

Reference list – full reference is now available for ref. 38.

Author Response

The authors present a nice review in which they address the roles of the INO80 and SWR1 chromatin remodeling complexes in plants, and compare the functions and subunit compositions of the SWR1/INO80-c of plants, yeasts and mammals. They also discuss the involvement of SWR1/INO80-c subunits in regulating plant growth, development and environmental responses.

I consider that the scientific topic being addressed is appropriate and interesting. The paper is well written and clear in all its sections. Cited references are appropriate and adequate, listing a large number of publications in high-impact journals.

I recommend the publication of this paper in the International Journal of Molecular Sciences, and have very few minor comments and suggestions listed below:

The role of ARP6 in maintaining genome stability and in DNA repair has been omitted from Table 2. Regarding this function, reference Rosa et al. 2013 could be used, together with ref. [49] mentioned in line 154.

Response: Thank you for your suggestion. We added this content in the revised manuscript (Table 2).

Li et al. (2005) have shown that in yeast the presence of Htz1 under the inactivated conditions is essential for optimal activation of a subset of genes, and suggest that Htz1 may serve to mark quiescent promoters for proper activation. They have also demonstrated that Htz1 is not generally responsible for nucleosome positioning, even at those promoters where he is highly enriched, and that incorporation of Htz1 into nucleosomes inhibits activities of histone modifiers associated with transcription. If found convenient, this information could be used to supplement the information brought in lines 76 - 84.

Response: Thank you for your suggestion. We added this content in the revised manuscript (line 137-141).

The abbreviation description for FLC is brought in line 168, although the first appearance of this term is in line 116. In addition, some terms throughout the text appear without their abbreviation explanation, e.g. HSA (line 144), GI (line 218), etc. Due to the quite extensive number of abbreviations used in the text, maybe abbreviation list could be provided at the end of the manuscript, if possible.

Response: Thank you for your suggestion. We added the abbreviation list in the revised manuscript (line 474).

Term Htz1 (used in line 93 and 98) has not been brought into association with H2A.Z in the manuscript.

Response: Thank you for your suggestion. We added details in in the revised manuscript (line 137). At the end of the line 91, authors could refer also to Table 1, in addition to the existing references.

Response: Thank you for your suggestion. We added details in in the revised manuscript (line 159). Figure numeration is missing from the Figure 1 description.

Response: The figure number was added in the revised manuscript (line 823).

Reference list – full reference is now available for ref. 38.

Response: We revised the reference in the revised manuscript (line 625).

Reviewer 2 Report

Dear Authors, I consider your work as a very well organized and written review. 

I have just few suggestions for you to improve the overall quality.

Line 51: Change the purple colour of the table that is unreadable, or use no colors that is even better.

Line 68 to 71: please write the concept in different way. It results hard to comprehend immediately.

Line 80 to 82: same as before, please try to rewrite the sentence to better clarify to readers.

Line 125: reference is written extended, please put the number and no ref extension.

Line 158: there is no figure number indicated. Please add it.

Figure 1: figure is readable but you have space to enlarge it a little bit. Please do.

Author Response

Comments and Suggestions for Authors

Dear Authors, I consider your work as a very well organized and written review.

I have just few suggestions for you to improve the overall quality.

Line 51: Change the purple colour of the table that is unreadable, or use no colors that is even better.

Response: Thank you for your suggestion. We revised the table 1 (line 513).

Line 68 to 71: please write the concept in different way. It results hard to comprehend immediately.

Response: Thank you for your suggestion. We rewrote these sentences in the revised manuscript (line 126-132).

Line 80 to 82: same as before, please try to rewrite the sentence to better clarify to readers. Response: Thank you for your suggestion. We rewrote these sentences in the revised manuscript (line 146-148). Line 125: reference is written extended, please put the number and no ref extension.

Response: Thank you for your suggestion. We changed this reference in the revised manuscript (line 201).

Line 158: there is no figure number indicated. Please add it.

Response: The figure number was added in the revised manuscript (line 823).

Figure 1: figure is readable but you have space to enlarge it a little bit. Please do.

Response: Thank you for your suggestion. We redo the Figure 1 in the revised manuscript (Figure 1, line 829).